# Optimization of Clustering in Wireless Sensor Networks: Techniques and Protocols

**Ahmed Mahdi Jubair [1], Rosilah Hassan [1,\*], Azana Hafizah Mohd Aman [1], Hasimi Sallehudin [1], Zeyad Ghaleb Al-Mekhlafi [2], Badiea Abdulkarem Mohammed [3] and Mohammad Salih Alsaffar [2]**

[1] Center for Cyber Security, Faculty of Information Science and Technology, Universiti Kebangsaan Malaysia, Bangi 43600, Malaysia; p94660@siswa.ukm.edu.my (A.M.J.); azana@ukm.edu.my (A.H.M.A.); hasimi@ukm.edu.my (H.S.)

[2] Department of Information and Computer Science, College of Computer Science and Engineering, University of Ha'il, Ha'il 81481, Saudi Arabia; z.almekhlafi@uoh.edu.sa (Z.G.A.-M.); m.alsaffar@uoh.edu.sa (M.S.A.)

[3] Department of Computer Engineering, College of Computer Science and Engineering, University of Ha'il, Ha'il 81481, Saudi Arabia; b.alshaibani@uoh.edu.sa

\* Correspondence: rosilah@ukm.edu.my

**Abstract:** Recently, Wireless Sensor Network (WSN) technology has emerged extensively. This began with the deployment of small-scale WSNs and progressed to that of larger-scale and Internet of Things-based WSNs, focusing more on energy conservation. Network clustering is one of the ways to improve the energy efficiency of WSNs. Network clustering is a process of partitioning nodes into several clusters before selecting some nodes, which are called the Cluster Heads (CHs). The role of the regular nodes in a clustered WSN is to sense the environment and transmit the sensed data to the selected head node; this CH gathers the data for onward forwarding to the Base Station. Advantages of clustering nodes in WSNs include high callability, reduced routing delay, and increased energy efficiency. This article presents a state-of-the-art review of the available optimization techniques, beginning with the fundamentals of clustering and followed by clustering process optimization, to classifying the existing clustering protocols in WSNs. The current clustering approaches are categorized into meta-heuristic, fuzzy logic, and hybrid based on the network organization and adopted clustering management techniques. To determine clustering protocols' competency, we compared the features and parameters of the clustering and examined the objectives, benefits, and key features of various clustering optimization methods.

**Keywords:** clustering; optimization; meta-heuristic; fuzzy logic; wireless sensor network

## 1. Introduction

The past few years have witnessed much research interest in Wireless Sensor Networks (WSNs). This increasing interest has demanded a comprehensive study that gives researchers a solid understanding of this field of research. A WSN is an ad hoc network consisting of a few sensor devices that cooperate to bring about particular functions, such as sensing a physical environment, making decisions, and transferring the sensed data to an appropriate end. Since the development of WSN technology, it has been a vital component of the Internet of Things (IoT) by providing a platform for connecting numerous devices and sharing information among these devices to improve user control of the environment.

There are four basic components of each WSN sensor node: transceivers, sensors, the power supply, and microcontrollers. The work of the sensors is to measure the relevant parameters in real-time, while the processing unit processes the sensed parameters and forwards them to the Base Station (BS) via the communicating unit using a single hop or

intermediate nodes [1]. WSNs have found applications in real-time monitoring activities, such as military surveillance, health monitoring, agriculture, disaster management, and more [2,3]. The deployment of WSNs is mainly in areas that may not be easily reachable by humans. An interesting research aspect of WSN is related to energy usage and balance in the network.

The limited and non-rechargeable nature of node power supplies has driven research into new ways of improving the energy balance and energy efficiency in WSNs [4]. The lifetimes of sensor batteries are limited, and efforts are being made to enhance these sensors' service lives by designing energy-efficient routing protocols. Routing is a tedious task in a WSN, as it is the basic feature that differentiates WSNs from other ad hoc wireless networks. Energy-efficient routing methods are necessary for a WSN to transfer sensed data from the Sensor Node (SN) to the BS; this will improve the service life of the network. Sensor nodes in WSNs are normally grouped into clusters, and this clustering method is used in WSNs to ensure the scalability of the network. It also guarantees efficient resource use and management of limited network resources, saving energy and conserving the stability of the network [5].

Clustering schemes are deployed in sensor networks to ensure efficient resource usage and reduce communication overheads, reducing the system's overall energy usage and keeping interference low among the SNs [6]. According to [7], clustering routing is mainly deployed to reduce the data transmission rate via the mechanism of information pooling in the Cluster Head (CH). This mechanism reduces communication-related energy usage, hence decreasing energy demand by the SN. Another reason for deploying clustering is to improve load balancing, thereby prolonging the service life of the network. Clustering strives towards improving the network lifetime by ensuring balance in the duties of the CHs.

Most types of WSN rely on cluster-based protocols to reduce energy consumption by the SNs. These clustering techniques used in WSNs are based on several optimization methods for efficient handling of clustering operations. To understand the existing problems in this research area (WSN clustering), conducting a literature review seems essential for providing a more profound knowledge of the different clustering methods and their limitations.

This review aims to examine the existing clustering techniques used to improve the performance of sensor networks based on different design characteristics and optimization methods. This article will help equip researchers with better knowledge of clustering-based optimization techniques, their classification and function based on the applied optimization methods, and an understanding of the basic limitations of the existing clustering techniques.

This study also focuses on clustering protocols in WSNs from the perspective of optimization algorithms. Recent studies on optimized clustering solutions were extensively reviewed and examined in detail for protocols regarding the methodology and properties of the considered algorithms. These techniques were evaluated based on the clustering standard parameters and optimization process parameters used with respect to their compatibility with different network characteristics in order to compare cluster protocols. Optimization parameters were used in this review for a better evaluation of the techniques and a general understanding of the clustering protocols. The parameters are presented considering various classes of optimization techniques, such as meta-heuristic, fuzzy and hybrid algorithms. The following are our major contributions:

- Provision of a novel perspective and method for conducting a review of the existing optimization techniques for clustering protocols;
- Provision of a novel optimization algorithm-based classification method;
- Provision of a comprehensive review and evaluation of the available literature based on clustering parameters and optimization for WSNs to understand the protocols and their related methodologies.

The remainder of the article is organized as follows. Section 2 describes how the review was conducted. Section 3 presents the existing literature surveys, followed by Section 4, which discusses clustering fundamentals. Section 5 provides an overview of Clustering Process Optimizations. In Section 6, clustering approaches are classified into different categories, and a summary of each clustering technique is presented in order to highlight the objectives and evaluation functions. Moreover, the section provides tables comparing the protocols that are examined and discussed with respect to classification. Finally, Section 7 details the conclusions.

## 2. Literature Research Process

This article adopted a mixture of optimization techniques in order to review the available literature on WSN clustering protocols. According to [8], this method has advantages over the narrative style. It can identify areas covered by existing studies and highlight gaps, approach literature from various perspectives and promote new insights. This survey of optimization techniques and clustering protocols used an online database and other resources to find all articles that met specific criteria, entered information concerning each study into a personal database, and summarized the current state of the table. The comprehensive literature review process is summarized in Figure 1.

- Search query: a generic search query was constructed for the purpose of search uniformity. This generic search query was utilized when searching for studies within our data sources, including terms "Clustering", "Clustering Protocols", "Optimization", "Techniques", "Wireless Sensors Network," and "WSN"; the query used for each data source was highlighted in detail. Accordingly, all the search terms were consistent. With this, we conducted a uniform search on all the data sources. However, each database had unique interfaces for advanced search with connectors such as OR and AND sometimes being switched depending on the data source used;
- Data sources: four data sources were used, which were IEEE Xplore, Science Direct, Springer Link, and ACM. By utilizing these data sources, all relevant works in the field of research were expected to be retrieved. This study considered these data sources to be the key sources for obtaining any possible related works;
- Time period: the search within each data source was set to retrieve only studies dated from 2010 to 2021. This was done to ensure that up-to-date studies were the only ones included. Additionally, earlier cited studies were included, as long as the study's full text was available;
- Applying exclusion criteria: our research focused on academic articles published in English. We also considered news articles, books, and annual reports touching on optimization clustering protocols and techniques of WSN;
- Data extraction: each paper was recorded based on author of record, year of publication and the journal in which the study was published. Subsequently, each article was classified according to the method used and whether the analysis covered state-of-the-art WSN clustering optimization protocols/techniques;
- Identifying data synthesis: analysis was performed to identify the optimization of WSN clustering protocols/techniques and recommend future studies.

For this paper, all scientific papers were accumulated from online resources. Digital databases including IEEE Xplore, Science Direct, Springer Link, and ACM Digital Library were used to obtain scientific articles for this survey. The literature was chosen based on the following keywords used for the search process: optimization of clustering protocols in WSN, clustering in WSN, clustering protocols in WSN, principles, requirements, and challenges. The search was conducted in April 2021 and was based on article titles. Twenty-three studies were selected from the data sources based on their relevance to the research topic and reviewed.

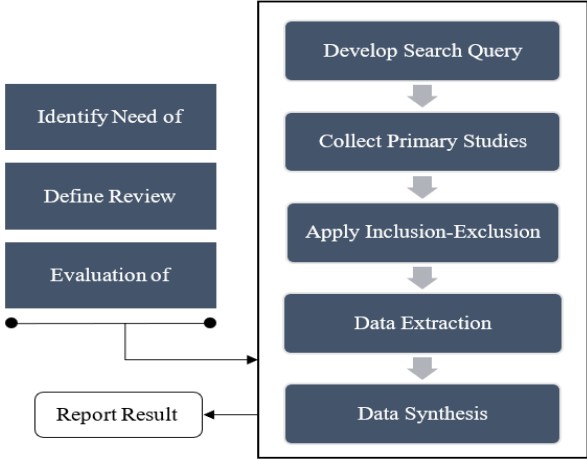

**Figure 1.** Literature review process.

### 3. Existing Literature Reviews on Clustering Based on Optimization

Previous scholars have extensively reviewed and surveyed clustering techniques in WSNs. Table 1 summarizes these studies based on their contributions. Ref. [9] was the first attempt to survey swarm intelligence-based routing techniques by considering their application areas and simulation platforms. However, only swarm intelligence-based protocols were considered in this survey, while other promising swarm-based protocols were not considered.

The cluster-based routing techniques for homogeneous SNs were comprehensively reviewed in Ref. [10] by classifying them based on their objectives and clustering methods; the considered parameters of the review were CH selection, data aggregation, cluster formation, and data communication. In each phase, detailed classifications of the clustering techniques for homogeneous networks were also provided with reference to existing studies since 2011. The reviewed CH selection methods were classified into assisted, multi-factor, and self-organized evaluation schemes.

Ref. [11] reviewed the existing clustering routing protocols by classifying clustering algorithms into two categories (data-transmission and cluster-construction routing techniques). Sixteen popular and important clustering methods were considered in this review, while other new approaches, such as fuzzy and evolutionary-based methods, were not considered.

Fuzzy modeling-based node clustering methods in WSNs were reviewed in this work, focusing on their advantages and limitations. A classification of fuzzy and hybrid fuzzy-based clustering methods was also presented.

Ref. [12] focused on cluster-based routing techniques based on various methodologies. The review focused on the positives and limitations of these techniques by classifying them into block, chain, and grid-based techniques. The methods were evaluated based on their scalability, cluster stability, delivery delay, and energy efficiency.

Clustering protocols were also reviewed in Ref. [13] based on their positives and shortcomings. Cluster-based routing techniques were classified and presented in three broad categories: block, grid, and chain-based clustering techniques. Existing schemes were also comparatively evaluated regarding delivery delay, energy awareness, load balancing, cluster stability, and algorithmic complexity.

The classification of different WSN clustering protocols into homogeneous and heterogeneous WSNs was presented in Ref. [14] considering the node and resource capabilities of the networks. Each protocol was reviewed based on its challenges, while the comparison included but was not limited to cluster count, inter-cluster communication, number of CHs, clustering objects, and complexity.

Unequal clustering techniques were surveyed in Ref. [15] based on their objectives and features. Unequal clustering techniques were categorized into probabilistic, preset, and deterministic methods and compared with respect to clustering/cluster properties and the clustering process. Some of the methods were also simulated to determine their energy usage and service life.

Existing clustering techniques were reviewed in Ref. [16] based on their general classification parameters and criteria; the clustering schemes were categorized into classical, fuzzy-based, meta-heuristic and hybrid meta-heuristic-based algorithms. Cluster-based routing protocols were also broadly categorized into methodology-based parameters and clustering-based parameters.

Ref. [17] presented ongoing machine learning or computational intelligence progressive methods. Computations were grouped based on their different computational intelligence uses into categories of swarm intelligence, fuzzy logic, neural network, genetic algorithm, and reinforcement learning. These computational intelligence uses were analyzed based on their scalability, data delivery rate, and data aggregation. We correspondingly noted that these methods improved the lifetime and service quality of the network. The hybrid model combinations also improved the interference of the network.

**Table 1.** Literature reviews of clustering protocols.

| Ref | Year | Area of Study | Contribution |
|---|---|---|---|
| [9] | 2011 | Swarm Intelligence | Categorized WSN routing protocols based on the concept of SI and its importance in routing |
| [10] | 2012 | Classical and Fuzzy-logic | Provided a classification of cluster-based methods based on their strategies and goals |
| [11] | 2012 | Classical | Review and summarized the goals of different clustering routing protocols. Provided a classification of WSN clustering techniques based on their cluster attributes |
| [12] | 2014 | Classical | Reviewed different clustering approaches by comparing their cluster size, complexity, algorithmic complexity, and cluster count |
| [13] | 2015 | Classical and heuristic | Reviewed the existing clustering methods in terms of their advantages and challenges, and classified the cluster-based routing methods into block, grid, and chain-based clustering |
| [14] | 2016 | Classical | Analyzed the popular clustering schemes quantitatively and qualitatively using performance metrics, such as cluster formation, communication, management, and complexity |
| [9] | 2018 | Classical, Swarm Intelligence | Compared existing homogeneous and heterogeneous clustering methods, as well as distributed and centralized clustering methods |
| [15] | 2019 | Classical, Fuzzy and Heuristic-based | Systematically analyzed the objectives, advantages, and challenges of some unequal clustering methods. The approaches were also classified and compared based on cluster properties, clustering process, and CH attributes |
| [16] | 2019 | Classical, Fuzzy and Heuristic-based | Considered classification criteria and parameters to evaluate some existing clustering methods. Four categories of clustering techniques were recognized in this work, which were classical, fuzzy-based, meta-heuristics-based, and hybrid meta-heuristics-based schemes |
| [17] | 2019 | Heuristic, fuzzy and machine learning | Review of machine learning-based hierarchical clustering techniques and classification of the algorithms based on their computational intelligence into swarm intelligence, fuzzy logic, neural network, genetic algorithm, and reinforcement learning |

## 4. The Fundamentals of Clustering

The three modes of SNs are sensing, computing, and communication, and these are energy-intensive processes. Technically, the level of energy required by the processor to transfer one bit of data is equivalent to the energy needed to compute several arithmetic operations. Furthermore, the physical environment of a heavily deployed SN network can generate a similar data rate in almost all the SNs, and the transmission of such data is redundant. Hence, it is crucial to merge all the factors that encourage the clustering of SNs

in an intelligent manner that allows for the transmission of compact data only (this is referred to as clustering).

In WSNs, clustering effectively manages network problems related to service life and energy use [18,19]. Clustering ensures energy conservation by adopting low-cost communication techniques [20] when it divides the network into different groups of nodes, called clusters. Each cluster has a CH that oversees the activities of the other nodes in the cluster [21]. The CHs can also reach the BS by creating a group and communicating with the BS in a multi-hop pattern. The CH first gathers the data collected by all the nodes before forwarding them to the BS [22]. The schematic of a clustered network in a WSN is shown in Figure 2. The designation of CHs in the clusters removes the problem of redundancy and reduces network energy consumption [23]. Different clustering techniques depend on various procedures and methods when executing clustering activities [24,25].

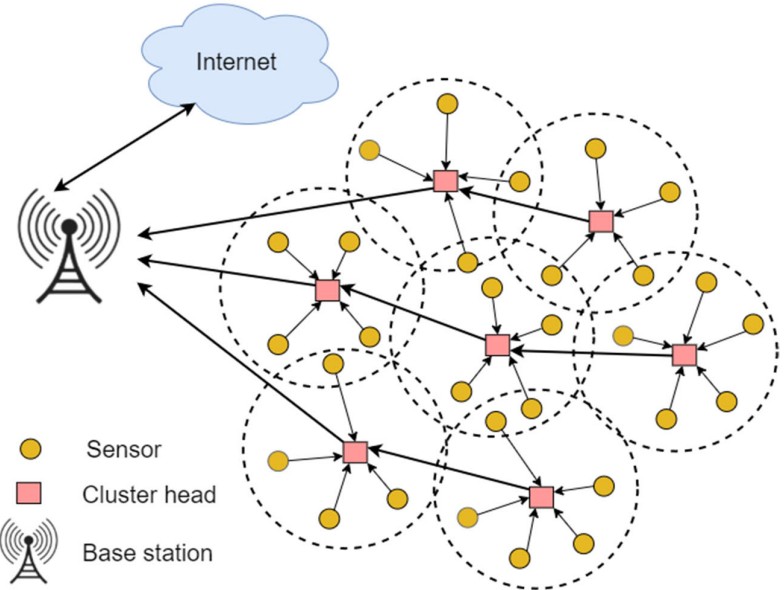

**Figure 2.** Clustering architecture in a WSN.

### 4.1. General Framework

In a WSN, clustering aims to maximize the service life of the network, which is carried out in two major phases; namely, cluster formation and CH selection.

### 4.1.1. Cluster Formation

This phase aims to reduce the workload of the CHs near the SNs by ensuring that each cluster has few members. Each member node is assigned to the nearest CH, based on the Received Signal Strength Indication (RSSI) [26]. The membership of each node within a cluster is determined by the received local data that the CH sends to nodes within its radius [27,28].

### 4.1.2. Cluster Head Selection

Practically, a CH is mainly used for information aggregation and distribution to the SNs; the selection of a CH is significantly crucial for energy usage optimization [29], meaning that efficient CH selection can improve the service life of the network [30]. Energy usage by CHs near the BS is usually higher in cluster-based networks, resulting in hotspot problems. This problem is addressed using unequal clustering algorithms [31]. The conditions that are considered during CH selection mainly include the CH's mobility, communication, and role. The metrics used during CH selection include the distance from the

CH to the nodes, the distance between the CH and the BS, the distances between the nodes and the BS, residual node energy, RSSI, node degree, cluster density, node weight, and position metrics. The CH selection communication criteria are determined by the data transfer rate from the cluster members to the CHs and from the CHs to the BS.

The nodes communicate directly with the CHs and the CHs communicate directly with the BS; this communication is established in single-hop or multi-hop modes. Another factor considered during CH selection in a WSN is CH mobility, because of the mobility of most WSN applications [32]. Fixing the CH entails that the clusters will remain fixed, which improves inter- and intra-cluster network management. By contrast, a mobile CH continually changes the sensors' cluster membership, requiring continuous monitoring of the clusters. The CHs sometimes move within limited distances to improve network efficiency through self-repositioning. The CH serves as a relay for the generated traffic and aggregates the data from the SNs. It can also sometimes serve as an SN when its action is based on the intended targets. CHs are also involved in ensuring load balancing and energy efficiency in the network.

### 4.2. Clustering Characteristics

Some attributes of clustering methods rely on the internal cluster structure to categorize different clustering protocols. Figure 3 shows a set of these attributes and can be applied in various WSN clustering protocols. Some of each attribute's definitions and uses in every clustering technique are summarized.

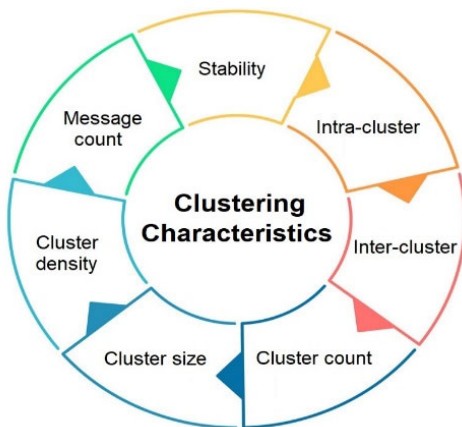

**Figure 3.** Clustering characteristics in WSNs.

- Inter-cluster head connectivity: Reflects the ability of SNs or CHs to establish communication with the BS. The clustering scheme must provide intermediate routing routes to the BS if the CH cannot establish long-distance communication;
- Cluster count: Refers to the number of clusters developed in each round; the higher the number of CHs, the smaller the cluster distribution size and the better the energy conservation. CH selection in some clustering methods is pre-assigned, meaning that the CHs can be randomly selected, resulting in different numbers of clusters;
- Cluster size: The optimum path length between the individual nodes and distance from the CH in a cluster. The smaller the cluster size, the better the energy usage, as the transmission distance and CH load are effectively reduced. The cluster size is fixed in some clustering methods, especially when clusters are fixed throughout their service life, but some clustering methods have a variable size for each cluster;
- Cluster density: Refers to the number of ordinary nodes in a cluster; reduction in energy usage by the CHs in dense clusters is a tedious task. Hence, most clustering methods rely on fixed clustering and opt for sparse cluster density (cluster density is variable for dynamic clustering methods);

- Message count: Refers to the required number of message transmissions for CH selection. The higher the message count, the more energy usage required for the CH selection procedure. Most non-probabilistic algorithms require message transmission for CH selection;
- Stability: Clustering schemes are adaptive if the members of a cluster are not fixed; otherwise, they are considered fixed because the cluster count cannot be varied during the process of CH selection. Fixing the cluster count improves the stability of an SN;

Clustered WSNs can be categorized into homogeneous and heterogeneous classes based on the type and function of the SNs in the network [33]. In the homogeneous class, all SNs exhibit similar features, processing, and hardware capabilities. The CH is normally rotated among the component SNs to ensure uniform energy usage within the network. On the other hand, the heterogeneous class contains two or more types of SNs: the first type is those with complex hardware and greater processing capabilities that are mainly used to create some sort of backbone inside the WSN [34], which are designed to serve as the CHs and can act as data collectors and data processing centers, while the second type is the participating sensors developed with lower capabilities, which are used to sense the desired parameters in the field.

Cluster formation remains the most interesting aspect of studies on clustering protocols; the other issues of concern are those relating to the number of clusters to be formed, the formation of the cluster members, and the size of each cluster. Energy consumption in sensor networks is believed to be dependent on the number of clusters, size of the clusters, and cluster density. Cluster size and cluster density are directly related to the cluster count, because the lower the cluster count, the more energy needed to manage the increased number of cluster members. Dense clusters affect the stability of the sensor network; it is important to increase the cluster count to reduce the cluster size and cluster density. One of the challenges in clustered WSNs is optimizing the cluster density, cluster count, and cluster size to improve the network stability and lifespan.

### 4.3. Solution Scope of Clustering

The clustering of nodes in a WSN is usually carried out with different purposes and objectives, but the commonest objective is energy conservation. These objectives can be categorized into primary and secondary categories; primary objectives are considered the most substantial and essential during the clustering process, while secondary objectives are considered less consequential but can be indirectly achieved through clustering the network nodes. An overview of the common objectives of clustering is presented in Figure 4. Some of the objectives of WSN clustering are briefly explained below.

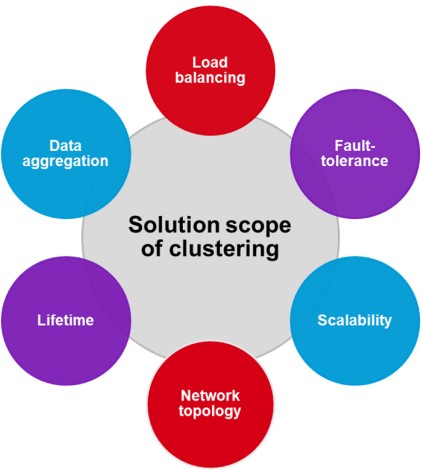

**Figure 4.** WSN clustering objectives.

1. Load balancing: Clustering is implemented to achieve low-energy intra-node communication and data processing [35]. Within the clustering architecture, the CHs perform the duty of data gathering and aggregation, long-range communication, and data forwarding. As such, the energy of the CHs deteriorates more quickly; this is why it is important to rotate these roles among all the network nodes using load balancing schemes to ensure energy efficiency;

2. Fault-tolerance: The deployment of WSN is usually in areas not easily accessible to human beings; therefore, such networks should have fault tolerance and be able to self-reconfigure in such deployment areas. The network must be designed such that the failure of one node cannot affect the general performance of the network [36]. According to Ref. [37], node clustering remains an effective way of making WSNs fault-tolerant and secure. The adaptive clustering method addresses faults in CHs through readjustments at the beginnings of pre-determined periods [14];

3. Scalability: The application area determines the number of SNs deployed in a sensor network [38]. The scalability of large networks can be increased by using hierarchical architectures where the network is divided first into virtual layers and subsequently into clusters [39]. In cases when a node in one cluster establishes a connection with another node outside its cluster, that node must have some information regarding the CH of the cluster of the node that intends to communicate with it; this increases network scalability and reduces the size of the routing table;

4. Network topology: The clustering of nodes into clusters makes it easy for CHs to manage location changes among the nodes within their clusters. Thus, it is more convenient for managing changes in network topology than flat architecture where there are numerous mobile nodes. Each CH in a clustered WSN is aware of its members' locations and levels of energy; therefore, the death of a node or its movement to another cluster is registered and reported instantly by the CHs;

5. Lifetime: The greatest challenge in WSNs, as mentioned earlier, is the extension of the network lifespan for as long as possible. It is believed that this objective can be achieved by deploying clustering mechanisms that meet all the highlighted characteristics. For instance, the positioning of CHs at the node center will ensure rotation of the CH role among all the nodes in the network and the effective utilization of sleeping schemes to improve the network's lifetime;

6. Data aggregation: Due to most data's uniformity in WSNs, data aggregation is important to prevent the transmission of similar packets through the network. Most data aggregation methods are signal processing-based; in WSNs, a standard data aggregation method combines all the incoming packets into one output packet [40]. All nodes must transmit their data to the BS in flat architectures via either a direct approach or a multi-hop one; however, some data aggregation methods can only be used in flat architectures that use data-centric applications [41]. Clustering allows data aggregation in the CHs, improving energy efficiency by reducing the total network load.

### 4.4. Recent Issues in Clustering

Communication is the most energy-intensive task in a WSN. Data communication and intra-node communication account for the highest rate of energy usage in WSNs. Some of the issues and challenges of WSNs with respect to communication include limited battery power, low computation ability, and limited bandwidth [42]. The major problems associated with most WSN clustering protocols are the heterogeneous nature of the nodes, the similar energy levels of all the SNs, and the uniform processing capability and memory of the nodes [43]. This problem does not occur in heterogeneous networks where the nodes are equipped with different bandwidths, energy sources, and processing and computational capabilities.

1. Energy: Computation and communication activities account for most of the energy usage of the SNs. Energy conservation improves the service life of the WSN due to the dependence of the network life on the battery life of the sensors.

2. Node deployment: This can be done either manually or randomly in a WSN. Manual node deployment involves using various deployment techniques to deploy the nodes manually. This form of deployment requires that the nodes follow a pre-determined path during routing [44]. In random node deployment, the sensors are randomly placed within a sensing environment in an ad-hoc manner.

3. Coverage: This is the physical space that can be covered by the deployed nodes; high coverage demonstrates the efficiency of the sensors for monitoring the target area. Connectivity denotes the ability of the nodes to initiate communication with the BS and the neighboring nodes. Network coverage and connectivity ensure the deployment of a sufficient number of nodes to monitor a given area.

4. Data aggregation: As SNs within a given region can sense similar parameters simultaneously, it is likely that they transmit similar data to the CH at the same time, thereby creating redundancy at the CHs [45]. The consequence is energy wastage, as the CHs will need much energy to process the excess data from the nodes, affecting network performance and lifetime.

5. Fault tolerance: Several factors can affect the functioning of SNs in the network, such as energy depletion, environmental and physical conditions, and so forth [46]. The failure of one node can affect the network performance. Hence, WSN protocols must be designed to be fault-tolerant and adaptable to environmental changes such that normalcy can be restored in case of failures.

6. Localization: This is a way of determining the positions of SNs in the network; most of the time, the position of SNs is determined by attaching GPS units to the SNs. However, this is a costly approach that cannot be deployed in all applications [47]. In WSNs, the deployed localization techniques strive towards finding the coordinates of the nodes using cost-efficient techniques.

7. Network dynamics: Most network applications are designed with static nodes, and in such a configuration, node movement is not possible after deployment. Some applications are designed with flexible nodes and BS; nodes in such networks can alter their positions to meet service demands. Mobile node routing is a complicated task due to the frequent changes in the route path and topology of the network.

## 5. Clustering Process Optimization

Clustering process optimization in WSNs comprises optimizing the processes of CH selection, data aggregation, cluster formation, and data communication. As shown in Figure 5, each component has some issues that need to be addressed. For instance, it is crucial to determine the optimal number of cluster formations, cluster density, and balance among the clusters during CH selection. In the cases of data aggregation and communication, both are closely related and must be maximized by determining the appropriate cluster size and degree of inter-cluster communication.

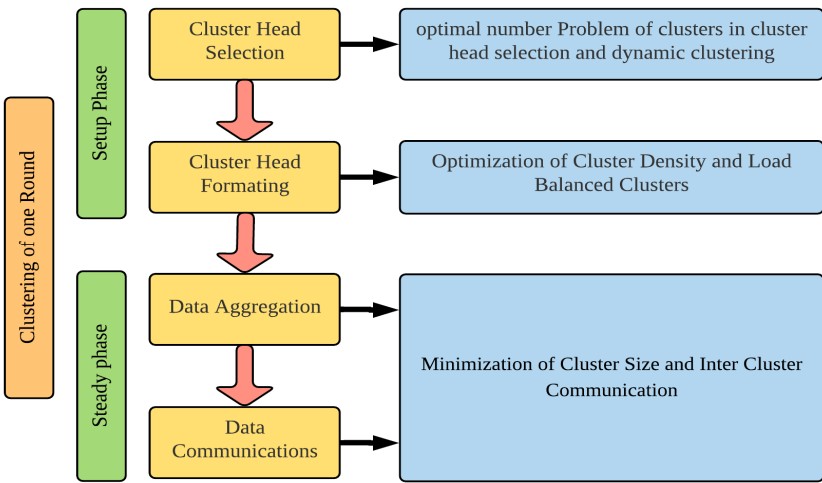

**Figure 5.** Clustering optimization methods in a WSN.

## 5.1. Cluster Head Selection Phase

Cluster head selection is the first phase of most clustering schemes, as the CHs are the gateway between the SNs and the BS. The role of the CH is to mediate communication between the SNs and the BS; hence, CH selection is a significant process for the subsequent clustering procedures to improve energy efficiency and lifespan of the network. Many studies have considered optimizing the CH selection process by implementing different techniques. These techniques are classified into self-organized schemes (distributed control) and assisted schemes (centralized control). Each SN in the self-organized scheme can execute its algorithm and decide on becoming the CH. By comparison, in the assisted scheme, the nodes are grouped by centralized authorities before selecting the CH for each cluster. Furthermore, self-organized schemes can be grouped into probability and non-probability-based schemes, while assisted schemes can be grouped into BS-based and CH-assisted schemes. The entire CH selection process is represented in Figure 6.

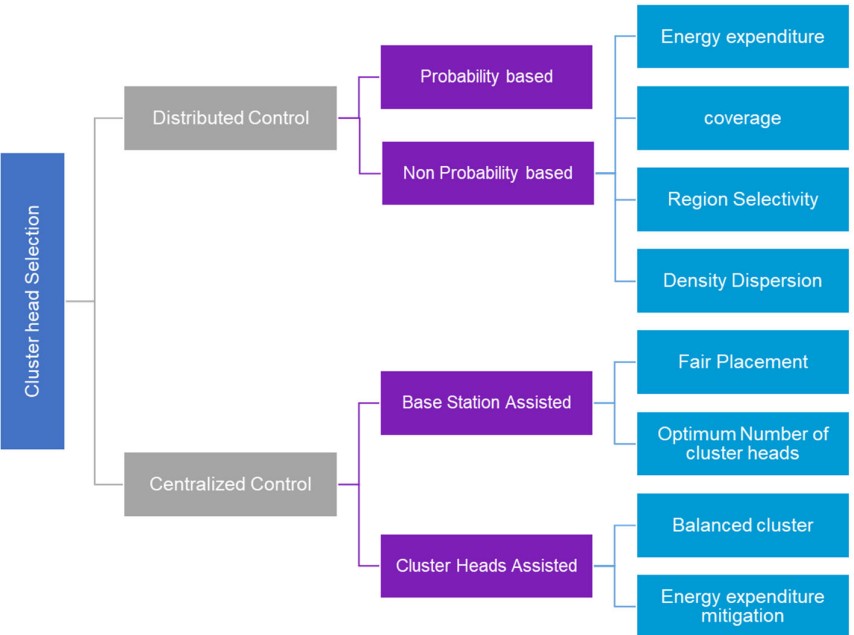

**Figure 6.** Cluster head selection process.

### 5.1.1. Probability-Based Clustering Optimization

Probability-based clustering schemes rely on the pre-assigned probability of each SN to determine the initial CH [48]. This pre-assigned probability is the primary criterion for determining the chances of selecting an individual node as a CH. The other conditions considered during CH selection are the residual energy, average network energy, initial energy, etc. These probability-based clustering algorithms ensure efficient energy usage and usually achieve faster convergence times and low packet exchange rates.

### 5.1.2. Non-Probability-Based Clustering Optimization

Non-probability-based clustering schemes consider more specific criteria for CH selection. These criteria depend on factors that can improve the performance of the CH selection schemes, such as density dispersion, energy expenditure, regional selectivity, and sensing coverage [49,50]. These protocols generally require more data exchanges and might sometimes lead to increased time consumption compared to random or probabilistic-based schemes. The energy consumption of the nodes selected as CHs is usually higher than that of the other nodes in the network; hence, the failure of a CH can cause data loss within the cluster member nodes [51]. The density dispersion-based clustering method selects the node with the highest residual energy as the CH [52]. Sensor nodes are deployed according to sensing coverage schemes to sense a given parameter while avoiding coverage gaps in the network [53]. Cluster head selection is not the first step in regional selectivity-based techniques. First, the sensors have to find their neighboring nodes in a specified radius, or perform an initial regional cluster formation considering the location of the nodes in the network, before selecting the most qualified node from each region as the CH using a distributed algorithm [54].

### 5.1.3. Base Station-Assisted Clustering Optimization

In BS-assisted clustering schemes, sensor nodes depend on the high processing power and inexhaustible energy resources of the BS by shifting the burden of the CH selection and cluster formation phases to the BS. Therefore, the end-user can control CH placement in the BS based on the network characteristics and the application type. However, these schemes require that the BS be periodically updated with relevant information by the SNs [55].

### 5.1.4. Cluster Head-Assisted Clustering Optimization

The cluster heads can collect updated information from the cluster members through regular communications in the data transmission stage. The CHs can rely on this information during the next round of CH selection to balance the clusters and avoid extra energy usage in the re-clustering stages. The schemes in this category are classified into balanced clusters and energy expenditure mitigation in re-clustering [56].

### *5.2. Cluster Formation Phase*

This phase begins with advertisement messages sent out by the newly elected CHs to announce their new status and ends with the join message sent by each node to its optimum CH. The cluster formation schemes are categorized into event-driven and optimal clustering schemes. Optimal clustering schemes strive to either manipulate the cluster size based on the application type and data transmission, or to balance and reduce energy usage by considering factors such as relay traffic, residual energy, and data correlation. Event-driven schemes strive to extend the network lifespan by removing dispensable clustering from the network and triggering the cluster formation stage only when necessary. Figure 7 illustrates the cluster formation process.

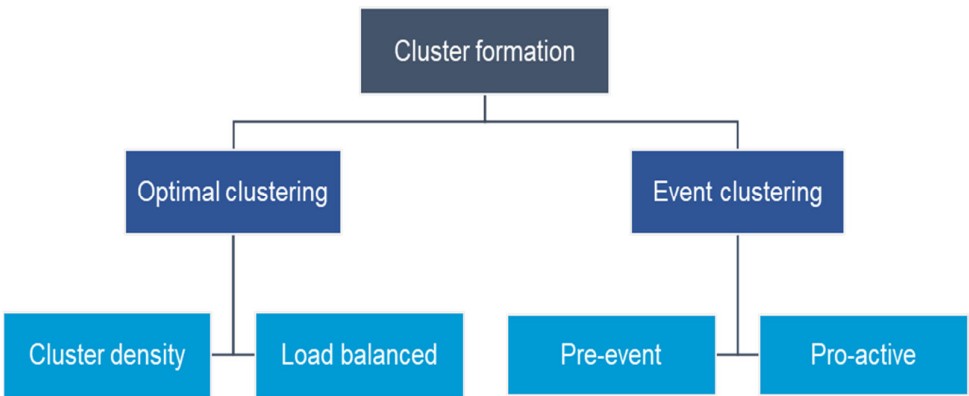

**Figure 7.** Cluster formation phase.

### 5.2.1. Optimal Clustering

In optimal clustering, cluster formation aims to reduce the rate of energy usage by the cluster members [57]. This process is achieved by assigning SNs to their nearest CHs after calculating their distances from the CHs based on the strength of the signals received as advertisement messages. In this clustering type, the size of the established clusters and the rate of energy usage within the clusters are not considered.

### 5.2.2. Event-Driven Clustering

In event-driven clustering, the focus is to generate energy-efficient clusters by preventing unnecessary and pro-active clustering [58]. However, significant overhead can result if clusters are formed in the entire field before the occurrence of an event; this overhead occurs in terms of network energy and processing and does not ensure better network performance in some applications [59].

### *5.3. Data Aggregation Phase*

Data from multiple sensors are gathered to remove redundancy during the transmission stage and provide fused information to the BS. Most data aggregation schemes aim at data gathering and aggregation in an energy-aware manner. Considering the limited energy of the SNs, allowing direct data transmission by all the SNs to the BS may not be energy-efficient. Additionally, the BS may not be capable of processing all the data generated by all the SNs. The data aggregation process is shown in Figure 8.

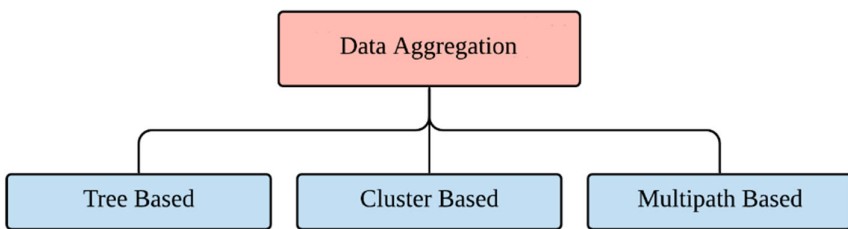

**Figure 8.** Process of the data aggregation phase.

### 5.3.1. Tree-Based Data Aggregation

These schemes are used for distributed data aggregation, and rely on some network data aggregator nodes; this ensures that the data paths of the SNs contain these data aggregator nodes [60]. The protocols in this category are used to construct an energy-aware data aggregation tree [61].

### 5.3.2. Cluster-Based Data Aggregation

This scheme depends on cluster formation; in every cluster, a CH serves as the point for data aggregation. An example of this method is the Low Energy Adaptive Clustering Hierarchy (LEACH) protocol [62,63], where CHs serve as the data aggregation points. The Hybrid Energy-Efficient Distributed (HEED) protocol is another cluster-based data aggregation protocol where the CH selection is based on the availability of multiple power levels at the SNs [64]. A combined metric takes into account both residual node energy and the nearness of the node to its neighbors.

### 5.3.3. Multipath-Based Data Aggregation

In these schemes, the aggregated data are divided by the SNs into several parts before being sent to a single point via numerous paths [65]. These schemes aim to improve network robustness by sending small duplicate data packets to the BS via multiple paths. Multipath-based data aggregation typically relies on a ring topology that allows the partitioning of the SNs into several levels with respect to their distance from the BS (number of hops) [6].

### *5.4. Data Communication Phase*

This phase involves transmitting the data aggregated by the CHs to the BS for further processing based on the application type. Packet transmission from SNs to the CH is called intra-cluster transmission, while packet transmission from the CHs to the BS is called inter-cluster transmission. Intra-cluster communication is sub-classified into single-hop and multi-hop transmission, while inter-cluster communication is sub-classified into direct and multi-hop transmission. All the SNs in a cluster can forward their sensed data directly to the CH during single-hop transmission [35], and this form of transmission does not consider distance [66]. Regarding multi-hop transmission, the sensed data obtained by the SNs of a cluster are first sent to the nearest neighbor for forwarding to the CH [67]. Cluster heads directly send the aggregated data to the BS in the direct communication method [68], irrespective of the distance. Multi-hop communication allows all the CHs to forward their aggregated data to the most proximal CH to the BS. Figure 9 presents the data communication process.

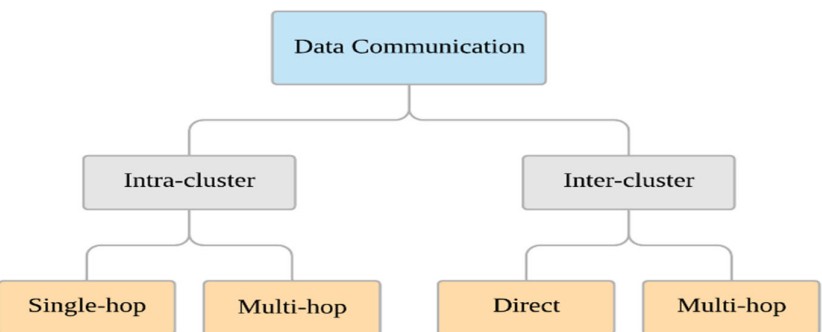

**Figure 9.** Data communication phase process.

### 6. Recent Advancements in Clustering Optimization Algorithms

Clustering is used in WSNs to meet performance requirements such as low energy usage. The design of clustering protocols does not only consider energy optimization, as there is also a need to ensure the quality of service (QoS) and a balance between numerous conflicting issues, such as service lifetime, coverage, and throughput [69]. Numerous bio-inspired, meta-heuristic, and artificial intelligence-based optimization techniques have been developed to address these concerns in the past decades. Energy efficiency is achieved in WSNs by combining optimization techniques with clustering protocols; this was accomplished by several researchers using different optimization techniques. In this

study, most of the currently used optimization algorithms in the WSN clustering process are classified into meta-heuristic-based, fuzzy-based, hybrid meta-heuristic-based, and hybrid fuzzy-based techniques, as seen in Figure 10.

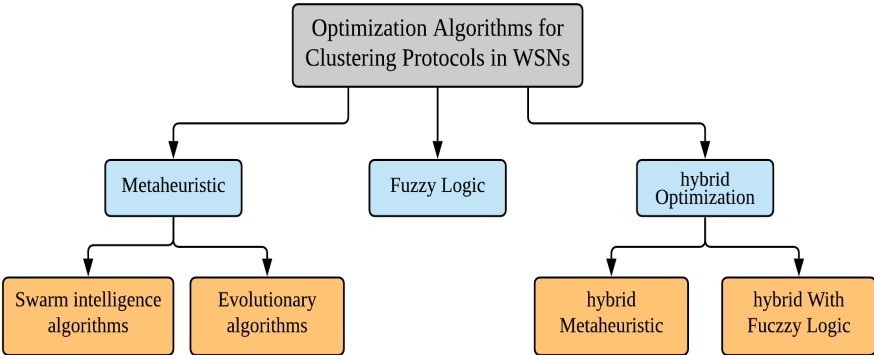

**Figure 10.** Classification of algorithms used in relevant articles.

The available research articles were categorized based on the optimization algorithms used, and a summary of each clustering protocol is presented to highlight the objectives and evaluation functions. This classification helps indicate the papers that discussed specific techniques, guiding beginners during research towards articles relevant for future studies. Detailed information concerning the selected articles is presented in Table 2.

**Table 2.** Overview of the selected articles.

| Ref | Protocol | Data Transmission | Cluster Topology | Cluster Size | Cluster Mobility | Deployment | Rotating the Role of CH |
|---|---|---|---|---|---|---|---|
| [70] | GATERP | One-Hop | Centralized | Equal | Static | Homogenous | Yes |
| [71] | HACH | One-Hop | Centralized | Unequal | Static | Homogenous | Yes |
| [72] | MRP-ACO | One-Hop | Hybrid | N/A | Static | Heterogeneous | N/A |
| [73] | GADA-LEACH | Multi-Hop | Distributed | Equal | Static | Heterogeneous | No |
| [74] | ERP | Multi-Hop | Distributed | Equal | Static | Heterogeneous | No |
| [75] | ABC-SD | Multi-Hop | Centralized | Unequal | Static | Homogenous | No |
| [76] | ICWAQ | One-Hop | Centralized | Equal | Static | Homogenous | No |
| [77] | Bee-Sensor-C | Multi-Hop | Distributed | Unequal | Static | Homogenous | No |
| [78] | TPSO-CR | Multi-Hop | Centralized | Equal | Static | Homogenous | No |
| [30] | PSO-ECHS | One-Hop | Centralized | Unequal | Mobile | Homogenous | Yes |
| [79] | DFCR | Multi-Hop | Distributed | Equal | Static | Heterogeneous | Yes |
| [80] | DFLBCHSA | Multi-Hop | Distributed | Equal | Static | Homogenous | Yes |
| [81] | MOFCA | Multi-Hop | Distributed | Equal | Static | Homogenous | Yes |
| [82] | FL-EEC/D | Multi-Hop | Distributed | Equal | Static | Homogenous | Yes |
| [83] | DFLC | Multi-Hop | Distributed | N/A | Static | Homogenous | Yes |
| [84] | HSA-PSO | Multi-Hop | Distributed | Unequal | Mobile | N/A | Yes |
| [85] | HABC-MBOA | Multi-Hop | Centralized | Unequal | Static | N/A | No |
| [86] | iCSHS | Multi-Hop | Centralized | Unequal | Static | Homogenous | No |
| [87] | hybrid GGWSO | Multi-Hop | Distributed | Equal | Static | Homogenous | Yes |

| [88] | DESA | Multi-Hop | Distributed | Equal | Static | Heterogeneous | No |
|---|---|---|---|---|---|---|---|
| [89] | GA-ANFIS | Multi-Hop | Centralized | Equal | Static | Homogenous | Yes |
| [90] | FAMACROW | Multi-Hop | Distributed | Equal | Static | Homogenous | Yes |
| [91] | LEACH-SF | One-Hop | Centralized | Equal | Static | Homogenous | Yes |

Comparison of the available techniques was based on qualitative metrics that portrayed the general features of the various protocols. The comparison of the clustering methods was based on the parameters for implementation, and these parameters portray the basic attributes of the related clustering methods, as discussed below. Figure 11 presents an overview of these implementation parameters.

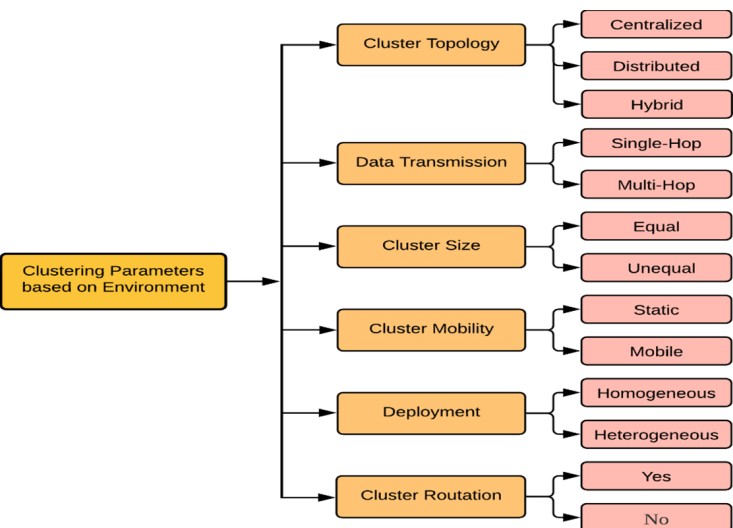

**Figure 11.** An overview of clustering parameters from the literature.

- Method: A clustering scheme can use either a centralized or a distributed method. Task implementation in these methods can be carried out using hybrid, distributed, or centralized mechanisms. The method's clustering phase can be distributed while the routing phase is centralized (either directly by a BS or with the help of a BS). The mechanism adopted across the entire algorithmic process is analyzed using this parameter;
- Data transmission: Some methods use a one-hop direct link between the CH and the SNs, while others rely on multi-hop connections. Multi-hop intra-cluster communication is suited for methods with few CHs and where the nodes are placed far from the CHs, or for methods where the SNs have transfer restrictions. This evaluation criterion considers the parameters as either one-hop or multi-hop;
- Cluster size: Cluster size can be equal or unequal based on the load distribution on the formed clusters. Load inequality among clusters is caused by variation in the distances between the BS and the SNs;
- Mobility: The CHs can be mobile or stationary; the movement of the mobile CHs is limited. The management of the topology of networks with mobile CHs is more tedious than that of those with stationary CHs;
- Deployment: WSN can be categorized into heterogeneous and homogeneous networks based on the resources of the SNs. SNs in the homogenous networks are equal in terms of energy level, computation power, and communication resources; CH selection is performed randomly or based on other criteria. In heterogeneous networks, the capabilities of the SNs are different. Therefore, CH selection is based on the specific capabilities of the SNs;

- Cluster Rotation: This criterion is used to determine the mechanism involved in the method to replace a CH. Certain methods replace their CHs periodically, while others replace their CHs after a pre-determined period or upon reaching a specified energy level. Each method tries to unify the network's energy usage level by adopting some energy threshold mechanisms.

*6.1. Meta-Heuristic*

The aim of developing optimization techniques is to provide solutions to NP-hard problems that cannot be addressed using traditional methods within a specified period. Meta-heuristics provide a global solution to such NP-hard problems, even though they sometimes fail to prove the best solution [92,93]. Combining clustering techniques with meta-heuristics is used to achieve optimal energy usage in WSNs, as they work together to identify the optimal solutions [94]. Researchers have used several meta-heuristics, including swarm intelligence, approximation algorithms, and evolutionary algorithm-based methods.

6.1.1. Evolutionary Algorithms

These algorithms are used for routing and clustering schemes; the Genetic Algorithm (GA) is the commonly used evolutionary algorithm for routing and clustering in WSNs. The GA is used to prolong the lifetime of CHs in order to improve network life [95] and network efficiency [96]. Different studies have presented hybrids of GA and Artificial Bee Colony (ABC) methods to enhance the clustering process, improve optimal routing through nodes, and improve the QoS [97,98]. This combination also reduced data energy usage in each round by reducing the total distance [99]. Despite the ability of the GA to solve multi-dimensional, non-continuous, non-parametrical, and non-differential problems, it still suffers from prolonged execution times, and cannot produce steady optimization response times in large populations. The protocols of clustering evolutionary algorithms-based approaches have been investigated using different parameters. to provide a general overview of the characteristics of their optimized Algorithms in clustering, Table 3 summarizes these studies based on their optimization methodology features.

- GATERP: GA-based threshold sensitive energy-efficient routing protocol

The GATERP proposed in Ref. [70] performed CH selection using GA parameters such as cohesion and cluster division. The system was equipped with an inter-cluster data transmission algorithm to elongate the network lifetime. Load balancing was improved by introducing a GA-based multi-hop communication mechanism that minimized energy usage. The GA was used to establish communication between the BS and the cluster heads; this aimed to achieve an optimal link cost for load balancing between distant CHs and reducing energy usage. The performance of the system was evaluated in terms of energy efficiency and energy utilization.

- HACH: Heuristic algorithm for clustering hierarchy protocol

The authors of Ref. [71] introduced the HACH protocol, which was developed for the selection of CHs and non-active nodes in each round using a stochastic sleep scheduling algorithm; this allowed SNs to enter a sleep state without disturbing the network performance. Energy usage and CH distribution in the WSN were also regulated by introducing a novel heuristic crossover operator. This system improved network performance by extending its lifespan under different energy heterogeneity settings.

- MRP-ACO: A multipath routing protocol

The multipath routing protocol (MRP) was developed [72] based on a combination of dynamic clustering and Ant Colony Optimization (ACO). The three phases of the algorithm are as follows: the initial phase is the selection of CHs based on their signal strength and residual energy; this step is aimed at prolonging the network lifespan. The second phase uses the ACO to establish the multiple paths between CHs and SNs. The multiple

paths were established with minimal energy cost using an improved ACO algorithm. The final phase is the selection by the CHs of dynamic routes for data transmission. In this algorithm, three types of ants were used: search ants (SANTs), backward ants, and abnormal ants. A new CH was selected whenever the current CH had residual energy of <50% of the average energy of the remaining nodes in that cluster with a stronger RSSI. The selected CH would initiate a new path discovery process when there were less than two multiple paths, implying a significant decline in the path reliability. The MRP was evaluated based on its energy usage, deviation energy, average energy, and network service time on a network with 100–500 nodes.

- GADA-LEACH: Genetic algorithm-based distance aware routing protocol

A GA-based distance-aware routing protocol for CH selection in WSNs was discussed in ref. [73]. CH selection was based on the residual energy of the CH and its distances from the BS and its neighbors. The selected CHs then initiated communication with the BS through a relay node. Each node in the sensor network is represented as a gene in a chromosome, whereas the length of the chromosomes is taken as equivalent to the overall number of nodes. The fitness of each chromosome is evaluated after establishing the initial population based on various metrics, such as the node energy, CH energy, and CH distance from the surrounding nodes and the BS. This algorithm considered the residual energy when reducing the number of CHs to ensure the selection of the most appropriate CHs. It also considered the chromosome length, equivalent to the overall number of nodes, which caused slow algorithmic convergence but had no regard for network-level CH disruption.

- ERP: A new evolutionary-based routing protocol

The evolutionary-based clustered routing protocol (ERP) was developed by Ref. [74] using a population set, divided into individuals, to increase the stability of cluster-based routing protocols, as it aims to select the optimal number of clusters. The protocol uses a binary tournament selection method where specific individuals are selected based on their fitness value from the population in the current generation. The ERP could extend the lifespan of the network and reduce energy usage, but it performed poorly in terms of stability awareness. The achieved protocol improved clustering by using evolutionary algorithms or meta-heuristics with modified fitness functions to confer compactness and take into account separation error.

**Table 3.** Comparison of studies Evolutionary algorithms.

| Ref | Algorithm | Optimization Methods | Objective | Optimization Process | Matrices | Simulation |
|---|---|---|---|---|---|---|
| [70] | GATERP | GA-based energy efficient threshold-sensitive protocol; GA-based protocol for CH selection with a novel fitness function and efficient encoding protocol | Network lifetime | Focus on GA-based identification of the nearest optimal path from each cluster to the BS, considering the distance and residual energy. | Energy consumption, network lifetime and stability | MATLAB |
| [71] | HACH | GA for sequential CH and inactive node selection per iteration | Network lifetime | Employed the GA and crossover to reduce the number of active nodes per iteration via switching some nodes into sleep or into inactive modes | Average energy, WSN heterogeneity, stability period and network lifetime | MATLAB |
| [72] | MRP-ACO | Proposed ACO-based load balancing for efficient traffic distribution over the already discovered multiple paths | Energy consumption | The probability model for CH dynamically chose a data transmission route that reduced energy utilization. | Average energy, energy consumption network lifetime | VC++ |

| [73] | GADA-LEACH | GA-based method for optimized CH and relay node selection for distance-aware routing | Network lifetime | Using GA for relay nodes acting as intermediaries between CHs and the BS | Network lifetime, throughput | MATLAB |
|------|------------|------|------|------|------|------|
| [74] | ERP | Inclusion of separation error and compactness criteria in the fitness function for guided searches for potential solutions | Network lifetime | Formulation of a novel fitness function that focused on the cohesion and separation error aspects of clustering | Network lifetime, death of first node, death of last node | MATLAB |

### 6.1.2. Swarm Intelligence Algorithms

The algorithms in this group are inspired by the natural life patterns of several animals or insects. Some of the algorithms in this group include Artificial Bee Colony (ABC), Grasshopper Optimization, Firefly Algorithm, Bacterial Foraging Optimization, and Particle Swarm Optimization (PSO). The swarm intelligence category as an optimization method shares two similar attributes with evolutionary computation techniques (i.e., population-based design and stochastic search). The following sub-section introduces swarm intelligence and provides the necessary background for understanding its basic concepts. The design of unorthodox optimization techniques in evolutionary computation is based on modeling or incorporating concepts and theories extracted from sociology, natural science, or psychology. An evolutionary algorithm, as earlier stated, mimics the concept of evolution by imitating the theories of genetics and natural selection in order to find the optimal solutions to optimization problems. Swarm Intelligence-based approaches are discussed here and compared by employing performance and features of optimization parameters to evaluate their working efficiency. These parameters aim to provide a general overview and comparison of these protocols. The features-based comparison is given in Table 4.

- ABC-SD: An energy-efficient cluster-based routing algorithm

The ABC-SD for cluster-based routing protocols was introduced in Ref. [5] for energy efficiency in WSNs; this new approach achieved low power consumption by exploiting ABC's search features (foraging behavior). The centralized control mechanism was also used to implement an LP formulation that relied on a multi-objective fitness function within the BS. The clusters were built by exploiting the efficient features of ABC, while communication between clusters was achieved based on the choice of CHs during clustering. The cost-based measure for optimal routing path selection was based on the energy–hop count balance; hence, it is a distributed approach. Comparison of the ABC-SD technique with other methods (such as PSO-C, LEACH-C, LEACH, ABC-C) in different network sizes showed that the protocol achieved better efficiency in coverage, network lifetime, and packet delivery ratio.

- ICWAQ: Improved version of cluster based WSN ABC Quality

Ref. [76]'s proposed algorithm, ICWAQ, is a system that relies on a message transferring technique to determine the distance between sensors in a WSN. This information is sent to the BS for the selection of CHs using an ABC algorithm. The choice of CHs is communicated to the network for the other nodes to find their closest CH based on signal strength. ICWAQ was developed with the aim of reducing WSN energy consumption. The performance of the proposed system was evaluated in comparison with LEACH and found to be up to 70% more energy efficient when the position of the BS was close to the network edge. However, ICWAQ introduced a significant level of overhead in the set-up phase because of the huge volume of messages passing between the sensors.

- Bee-sensor-C: An energy-efficient and scalable multipath routing protocol

Ref. [77] introduced Bee-Sensor-C, an extension of the BeeSensor system (C stands for cluster); it is an energy-efficient protocol for multipath routing that depends on the foraging pattern and dynamic clustering of the bee swarm. It ensured balanced network energy usage by adopting an enhanced multipath construction strategy that minimized

routing overheads and improved network performance and scalability. It employed a dynamic clustering technique for energy usage reduction and service life improvement. The HiveHeader was introduced in this protocol, in addition to the parameters used in the BeeSensor protocol. Cluster formation is the first phase of the Bee-Sensor-C; in this phase, the HiveHeader in the hive becomes the CH node upon detecting an event within a given area. In this way, the nodes within the event area are activated, followed by an estimation of the specific perceived attributes that would allow nodes with the event details to join the cluster. Multipath construction is the second phase of the system, wherein the CH propagates data to the BS; here, the CH scouts for the ideal node foragers for the existing multipath in the cluster. The transmission stage is the final phase and occurs upon the arrival of the backward scout at the CH with a specific path identity, followed by recruitment of the foragers using the bee waggle dance. The proposed approach was evaluated under two different network scenarios with varying nodes, and energy consumption and packet delivery rate were compared with BeeSensor, IEEABR, and FF-Ant.

- TPSO-CR: Two-tier particle swarm optimization protocol for clustering and routing

Ref. [78] proposed a two-tier PSO for clustering and routing. TPSO-CR is a protocol that relies on a clustering scheme for optimal CH selection; the protocol ensured efficient energy usage and improved network coverage and transmission reliability. The development of the routing algorithm was based on a novel particle encoding scheme and fitness function for finding the optimal path to connect the CHs to the BS. TPSO-CR is centralized and relies on information received by the BS, similarly to PSO-HC. The BS computes the average energy level of the nodes, so that only nodes with an energy level higher than the average can be considered as CHs for each round. This stage ensures that only nodes with sufficient energy to act as CHs are selected. In the next phase, the BS runs the clustering algorithm to find the best K-CHs, then implements the routing algorithm to construct the optimal routing tree. TPSO-CR was evaluated in two different scenarios called WSN#1 (with homogenous nodes) and WSN#2 (with heterogeneous nodes). The performance was compared with LEACH, EHE-LEACH, EEHC, LEACH-C, PSO-C, and GA-C and found to achieve a smaller number of non-clustered nodes compared to the other protocols in both test scenarios. TPSO-CR also achieved the highest throughput and less energy consumption than LEACH, EEHC, and EHE-LEACH. However, it consumed comparable levels of energy to LEACH-C, PSO-C, and GA-C;

- PSO-ECHS: A PSO-based energy-efficient cluster head selection algorithm

PSO-ECHS, presented by Ref. [30], is a PSO-based energy-efficient algorithm for CH selection implemented in two phases: the CH election phase and the cluster formation phase. PSO is implemented in the CH election phase based on nodes' residual energy and distances to the BS. The nodes first transmit their location and residual energy information to the BS for the execution of the PSO algorithm. During CH selection, PSO-ECHS considers each particle represented by the coordinates of the SNs to be chosen as CH to be an optimal position for the CHs. The PSO-ECHS was evaluated in three different scenarios varying the position of the BS. The first scenario placed the BS in the middle of the study field, while the second scenario placed the BS at the top right end of the study field, and in the last scenario, the BS was located outside the study field. The number of nodes was varied during these experiments, and the performance was compared with that of LEACH, E-LEACH, LEACH-C, PSO-C, and LDC. The performance of PSO-ECHS was better than that of the other protocols when considering energy utilization and extension of network lifetime. PSO-ECHS also achieved the highest rate of packet delivery to the BS in the evaluated scenarios using different numbers (i.e., 300, 400, 500, or 700) of nodes.

**Table 4.** Comparison of studies involving swarm intelligence.

| Ref | Algo­rithm | Optimization Methods | Objec­tive | Optimization Process | Matrices | Simula­tion |
|---|---|---|---|---|---|---|
| [75] | ABC-SD | Cluster-based routing protocol using the ABD algorithm. For­mulation of the clustering prob­lem as linear programming | Energy con­sumption Cost-based Function | Proposed the exploitation of the nature-inspired search features of the ABC meta-heuristic for building low-power clusters and selecting CHs | Throughput, net­work coverage, en­ergy efficiency | N/A |
| [76] | IC-WAQ | Development of an ABC algo­rithm for networks that lack a global positioning system | Minimize energy | ICWAQ exploited the fast and efficient ABC algorithm search mechanism to optimize node clustering during CH selection to define the routing paths | Residual network energy, fitness func­tion | MATLAB |
| [77] | Bee-Sensor-C | Scalable multipath and energy-efficient routing protocol for WSNs that is based on dynamic clustering and mimics the bee foraging pattern | Network energy con­sumption | Modeled bee agents to suit the limited-energy nature of WSNs, to enable the construction of clusters near event sources and find better quality multiple paths | Energy efficiency, control overhead, packet delivery rate, latency, routing building time | JAVA |
| [78] | TPSO-CR | A novel routing protocol based on PSO with a new scheme for particle encoding to ensure com­plete routing tree solutions and multi-objective fitness function derivation | Maximize energy | The clustering and routing prob­lems were formulated as LP for a clustering protocol based on PSO to balance data transmis­sion reliability and energy-effi­cient network coverage | Level latency, con­sumed energy, throughput, PDR | OM­NET++ |
| [30] | PSO-ECHS | Algorithm for CH selection based on PSO with an efficient scheme for particle encoding and fitness function. Normal clusters join their CHs based on a derivable weight function | Energy efficiency | Formulated a CH selection prob­lem as LP and derived the weight function for cluster for­mation | Energy consump­tion, network life­time, packets re­ceived | MATLAB |

*6.2. Fuzzy Logic*

Fuzzy logic was invented as a mathematical discipline for expression of approximate human reasoning. It allows for a measure of uncertainty or imprecision associated with the use of linguistic variables such as "most, many, frequently" through rules within a set called a fuzzy set [100]. In this section, fuzzy logic techniques were discussed and evalu­ated based on the fuzzy logic standard parameters and optimization process used (meth­odology and feature). To showcase a general outline of the different approaches, the com­parison of techniques presented in Table 5.

- DFCR: Distributed fuzzy approach to unequal clustering and routing algorithm

The DFCR [79] was presented as an algorithm with four major steps: sharing of in­formation, formation of clusters, formation of the virtual backbone, and routing of data. During the information sharing phase, the distances of each node from the BS and its neighbors are disseminated. Subsequently, the cluster formation step requires each node, in a distributed manner, to decide whether to become a CH by calculating its cluster ra­dius using the local information. Each node in this algorithm computes its energy level and distance from the BS as the fuzzy input parameters for obtaining the competency function 1 (CF1) for selection as CH. At this point, each node's latency is calculated based on the obtained CF1. Upon expiration of the timer, the node radius is calculated based on

CF1 and the derived fuzzy output, considering the neighbor cost and density of the CF2 output. Hence, CF1 and CF2 are considered fuzzy inputs; the cluster radius is estimated as the fuzzy output within each node, and a node is expected to distribute a notification message if it becomes the CH. Nodes that receive the message before the expiration of the timer will cease competing to be the CH. Each node will now compute its connection cost to the new CH in order to become members of the cluster incurring the least cost. The CHs are classified in the virtual backbone formation stage, followed by determination of their levels. Lastly, the data routing stage involves each CH selecting a member from the existing low-level CHs from which it received messages based on the minimum cost function.

- DFLBCHSA: A distributed fuzzy clustering algorithm for a WSN with a mobile gateway

The multi-objective distributed fuzzy clustering (DFLBCHSA) approach was developed to reduce delay in delivering data packets [80]. This approach formulated a CH selection process that considered node location and general state information; this was based on the weighted linear combination approach, which served to decide the values of the node location and its general state. CH selection was based on a fuzzy method using seven fuzzy descriptors. The resulting mobile gateway node position was predicted using the linear prediction method to reduce the problem of control message overhead in the network. The inputs to the fuzzy system were the energy, mean distance, neighbor count, and gateway location relative to the nodes.

- MOFCA: multi-objective fuzzy clustering algorithm

The MOFCA approach [81] was presented for addressing the issue of hot spots and early energy exhaustion in WSNs. This algorithm relies on various fuzzy inputs, as every node selects a number between 0 and 1 per round. When the selected number is less than the TH (i.e., the ideal percentage of the number of CHs), the node may become a temporary CH that will consider the fuzzy inputs in order to determine the competitive radius using fuzzy logic. Information can be disseminated based on the maximum competitive radius and the pre-determined radii of the temporary CHs. The temporary CH that receives the transmitted information from the higher energy level transmitter will cease to compete. However, when the energy levels of the two nodes are equivalent, a comparison of their density parameters will be conducted, and the one with the higher density will become the CH. The three fuzzy input variables in MOFCA are distance from the SNs, residual energy, and node density. Being a distributed method, MOFCA uses local decisions to determine the node competition radius and selection of the potential and final CHs.

- FL-EEC/D: Energy-efficient fuzzy logic-based clustering technique for hierarchical routing protocol

This fuzzy approach to CH nomination was proposed for efficient energy distribution management among the sensors in a WSN [82]. The fuzzy system's role is to create an opportunity for every node in the network to be selected as a CH. The chance of a node being chosen as the CH is determined by the system inputs, such as battery power, distance to BS, location suitability, proximity of neighboring nodes, and node density. All network nodes are compared to determine which node can serve as CH, so the node with the highest chance value can become the CH. With this approach, the proposed fuzzy model can select the optimal set of CHs, but its problem is that it cannot balance the load of the SNs among the CHs. The system analysis showed that the network lifetime was extended compared to LEACH, and the system's performance was efficient in a homogeneous WSN, though it showed poor performance in WSNs overall.

- DFLC: A distributed fuzzy logic-based root selection algorithm

The DFLC was developed as a fuzzy logic-based clustering technique implemented in a distributed manner by the nodes in a WSN [83]. The protocol considers the network a tree where any node can become the BS, CH, member, child, or parent node. For efficient

selection of the CH, a fuzzy logic engine is run by each node using five input parameters (residual node energy, node distance to the neighboring nodes, node distance to the BS, node density, and number of hops). Only the necessary nodes with a better chance of being selected as CH are considered during the implementation of the fuzzy logic engine. DFLC ensures the network does not fail when any SN has experienced service failure due to energy depletion. The performance of the DFLC was tested in the NS2 simulator and assessed in terms of energy consumption, number of active nodes, service life of the network, and volume of messages received within five node networks (totaling 100, 200, 300, 400, and 500 nodes). The DFLC aimed to reduce the number of propagated messages, reducing energy consumption. Comparison with LEACH, ACAWT, FCH, and CHEF showed that DFLC performed better based on the evaluation metrics.

**Table 5.** Comparison of studies involving fuzzy logic.

| Ref | Algo-rithm | Optimization Methods | Objective | Optimization Process | Matrices | Simula-tion |
|---|---|---|---|---|---|---|
| [79] | DFCR | Fuzzy logic-based clustering protocol for CH selection and computation of the cluster radius; fuzzy logic is applied to handle different levels of system uncertainties | To improve the service life of the network | Formation of unequally sized clusters by the clustering algorithm; cluster radius is computed based on a distributed FL approach | Network lifetime, energy efficiency, number of live nodes | MATLAB |
| [80] | DFLB-CHSA | Development of an entirely distributed fuzzy logic-based system to determine the eligibility of each node for being selected as CH, based on two input factors and application of linear prediction | To reduce energy use and delays in data propagation | Partitioning of the network into sub-areas, followed by deployment of mobile gateways to establish communication between the CH and the BS | Number of dead nodes, remaining energy | OM-NET++ |
| [81] | MOFCA | Handling the uncertainties in WSNs using fuzzy logic; the algorithm considers residual energy levels and the distance to the BS | To remove hotspot problems and balance load | Selection of the final CHs based on the energy levels of the nominated CHs; the energy levels of the CHs are pre-determined via a probabilistic model. | Number of live nodes, energy depletion | MATLAB |
| [82] | FL-EEC/D | Fuzzy logic-based CH nomination and distribution control using adaptive separation, which is a fuzzy-based centralized clustering method for energy-efficient routing frameworks in WSNs | To extend the service life of the network and minimize energy use | Fuzzy logic-based clustering technique for CH selection; enforces a separation distance between the CHs for even CH distribution in the monitored area. | Network lifetime, energy efficiency, number of live nodes | Dot not |
| [83] | DFLC | CH selection using a distributed fuzzy logic engine algorithm wherein the roles of the root nodes are dynamically changed based on their residual energy levels | To reduce the number of control messages | Each node implemented a fuzzy logic engine that prevented the forwarding of messages from nodes with less probability and kept them from being chosen as the new root. | Fault-tolerance, energy efficiency, network lifetime | NS2 |

### 6.3. Hybrid Techniques

Hybrid optimization is a way of choosing the optimization algorithm to apply from a range of algorithms that implement the same type of optimization. It assumes that two or more algorithms have been implemented for the same optimization. Therefore, to select

the best algorithm to apply in any situation, hybrid optimizations rely on existing optimization techniques to achieve this task.

6.3.1. Hybrid Meta-Heuristics

Ref. [101] reported using a hybrid technique based on an improved GA and a binary ACO to achieve optimal coverage, reduce data redundancy, and optimize the multi-objective function by determining the least number of sensors required. The role of the ABC was to extend the performance of the WSN and achieve better exploration and exploitation during CH selection [102]. For the improved ABC, the goal was to obtain optimal CHs for WSNs [103] and find the shortest routing path [104]. The fractional Grasshopper Optimization Algorithm (Fractional-GOA) was utilized to reduce energy usage by implementing sleep/wake scheduling in the nodes [105]. The optimal solution is considered the solution that offers the maximum fitness value and is the considered solution for sensor activation in the Fractional-GOA. We provide a general overview of the characteristics of hybrid meta-heuristic techniques. Table 6 summarizes these studies, investigated using different parameters.

- HAS–PSO: Hybrid HSA and PSO algorithm for energy-efficient cluster head selection

In Ref. [84], the HAS–PSO algorithm was developed for CH selection based on the residual energy and distance to ensure better search efficiency and convergence. The first step in the has–PSO is to select the network parameters and assign values to the velocity, particle fitness function, and hybrid matrix. These parameters describe the Particle Harmony Memory (PHM) for the generation of an improved harmony. The hybrid system achieved throughput and residual energy values approximately 83.21% and 29.14% higher, respectively, than those of the component algorithms. The hybrid nature of HAS–PSO ensured a balance between exploitation and exploration capabilities, thereby contributing to energy efficiency. The system was suited for ensuring balanced energy efficiency for a prolonged network lifetime. Analysis showed that the hybrid HAS–PSO achieved better network lifetime, standard deviation, FND, and LND than LEACH, HSA, and PSO.

- HABC–MBOA: Hybrid Artificial Bee Colony and Monarchy Butterfly Optimization Algorithm

The HABC–MBOA was presented in Ref. [85] as a CH selection method based on ABC and ACO for effective clustering and improvement of network lifespan. The system was proposed to improve the balance between exploitation and exploration and prevent local search entrapment. The approach replaces the employee bee stage of ABC with the mutated butterfly adjusting operator of MBOA to avoid local minima entrapment. This new method addressed the issue of insufficiency in global search in the ABC method. The system achieved better throughput and residual energy values than some existing protocols, such as EPSOCHSS and FC-ABCAICHSS.

- iCSHS: Integrated clustering and routing protocol for WSN using Cuckoo and Harmony Search

This algorithm was proposed in Ref. [86]. The improved Cuckoo Search (CS) algorithm was developed to address the issue of WSN clustering; an improved Harmony Search (HS) was also incorporated into the protocol to address the routing problem. The improved HS served in the transfer of the aggregated data from the CH to the BS. The role of the improved CS was to establish the optimal set of CHs among the normal SNs, while the HS found the optimal tree that would connect the CHs to the BS. A problem with this method is that it did not balance the load of the SNs among the CHs. Hence, the CS–HS hybrid was evaluated based on the average power loss, path of inactive nodes, path of active nodes, and network lifespan. This algorithm balanced network lifespan with energy usage.

- Hybrid GGWSO: Hybrid model for security-aware cluster head selection

The hybrid GGWSO [87] was presented as a framework for improved data collection based on relevant data. The status of the gathered data is updated using the anchoring nodes while the overall gathering values are minimized to improve link capacity, capability, and data flow conservation. This protocol aimed to improve network lifetime by ensuring that the algorithm addressed some of the salient issues related to energy, delivery delay, security, and distance. The equalization procedure was used to partition the entire dataset into smaller components. The performance of the GGWSO was compared with that of various traditional methods such as fractional ABC, GWO-based CHS, GSO, and ABC.

- DESA: Lifetime improvement using hybrid differential evolution in WSNs

The DESA protocol was presented by [88] as an integrated SA–DE protocol for CH selection during WSN clustering. This protocol aimed to prevent the earlier failure of CHs to achieve an extended network lifetime. The four phases of the proposed DESA are population initialization, application of the mutation operation, crossover operation, and selection of the next generation. After the population is randomly initialized, the opposite point technique generates another population set known as the opposite population. Then, number of fittest individuals are selected from the opposite population set for the current generation. The selected mutation strategy is determined by the value of the chosen random number; if the selected random number is greater than the threshold value, DE/rand/1 will be performed. If not, DE/current-to-best/1 will be performed. The new approach outperformed LEACH, HSA, MHSA, and DE by 70%, 50%, 40%, and 60%, respectively.

**Table 6.** Comparison of studies involving hybrid meta-heuristic techniques.

| Ref | Algo-rithm | Optimization Methods | Objec-tive | Optimization objectives | Optimization Process | Matrices | Simula-tion |
|---|---|---|---|---|---|---|---|
| [84] | HSA and PSO | The hybrid HSA–PSO allows the movement of particles from region to region via updating their velocity and position at the end of each iteration | Maxim-izing the network lifetime | Energy usage in and dis-tance between CHs | The hybrid approach makes use of the high searching efficiency of HAS combined with the dynamic nature of PSO | Number of live nodes, number of dead nodes, throughput and residual energy | MATLAB |
| [85] | HABC-MBOA | The employee bee phase of ABC was replaced with the mu-tated butterfly adjusting opera-tor of MBOA in the algorithm to prevent premature convergence and entrapment at the local op-timal point; this was achieved by ensuring a balance between exploitation and exploration | Maxim-izing the network lifetime | Number of sensor nodes, maximum number of rounds, di-mensions of sensor nodes | The ordinariness char-acteristics of ABC that permit the search pro-cess to move from one region to another were updated based on the position and velocity determined at each round of implementa-tion | Number of live nodes, throughput, residual en-ergy | MATLAB |
| [86] | iCSHS | Improved CS-based CH selec-tion protocol with a new multi-objective function with four pa-rameters | Maxim-izing the network lifetime | Residual node energy, degree of node, intra-cluster dis-tance and cov-erage ratio | Improved HS-based in-ter-cluster multi-hop routing protocol with a new fitness function | Energy con-sumption, net-work lifetime, number of dead nodes | MATLAB |

| | | | | | | |
|---|---|---|---|---|---|---|
| [87] | GGWSO | Formulated a new, efficient and objective function encoding technique for the selection of load-balanced CHs | Energy consumption | Build model of the cluster head focused on energy, distance, delivery delay, and security | Optimized energy resources and usage via organizing network nodes into clusters to improve the network lifetime | Energy consumption, network lifetime | MATLAB |
| [88] | DESA | DE-based local search together with SA for finding the global optima; this is aimed at improving the performance of WSNs using optimal CHs | Maximizing the network lifetime | Residual energy | The four phases of the DESA include population vector initialization, mutation, crossover, and selection of the next generation, as performed in the traditional DE algorithm | Residual energy, lifetime, throughput | MATLAB |

### 6.3.2. Hybrid Fuzzy

Hybrid optimization approaches are classified into fuzzy-based and meta-heuristic-based methods [106]. Some nodes do not assume CH status because only one CH can be selected within their transmission range. At the expiration of a timer, one CH is selected by every node based on the strength of the received signals from potential CHs. After data aggregation and cluster formation by the CHs, the data are propagated to the BS following certain rules. In this section, fuzzy hybrid techniques from various studies were discussed and evaluated regarding the methodologies and features. To illustrate a general outline of the different approaches, the comparison of techniques has been presented in Table 7.

- GA ANFIS: Increasing WSN Energy Efficiency to Choose a Cluster Head and Assess Routing

In Ref. [89], GA-ANFIS was proposed, based on the GA and ANFIS, for WSN clustering. A weighted trust evaluation method was proposed in this method for the discovery of dangerous nodes in the network. The GA is first applied to form clusters considering the distance between the SNs and the BS; then, a fuzzy logic system is applied to find each SN's chances of becoming the CH based on two fuzzy inputs (node energy and node distance from the BS). The last stage is applying the trust evaluation method to detect dangerous nodes and isolating them from the system.

- FAMACROW: Fuzzy and ACO Based Combined MAC, Routing, and Unequal Clustering Cross-Layer Protocol

FAMACROW [90] was presented as a cross-layer hierarchical protocol that combined fuzzy logic, ACO, and MAC for routing and unequal WSN clustering. This protocol performs CH selection, inter-cluster multi-hop routing, and unequal cluster formation. The three phases of FAMACROW include the setup phase, the neighbor finding phase, and the steady-state phase. The setup phase involves sorting the nodes into layers, while during the neighbor finding phase, the details of each node are transmitted using the non-persistent CSMA MAC protocol. The activities in the steady-state phase comprise CH selection, clustering, and data delivery. The selection of the CHs is implemented using fuzzy logic and ACO. The fuzzy logic is implemented using three input parameters for CH selection: residual energy, communication link quality, and number of neighboring nodes. The energy hole problem is solved by constructing unequal clustering structures in the protocol, forming smaller-sized clusters nearer to the BS. The inclusion of link quality in the CH selection process improved the reliability of the system.

- LEACH–SF: Optimized Sugeno fuzzy clustering algorithm

The LEACH–SF algorithm [91] is an adaptive fuzzy clustering technique that clusters sensor nodes into balanced clusters using the fuzzy c-means algorithm. At the same time, the appropriate CHs are selected using the Sugeno fuzzy inference system. It is an energy-efficient routing protocol that extends network service time through routing via the CHs of the WSN. Since tuning the fuzzy rules is the most crucial issue in this framework, the Sugeno fuzzy system relies on local sensor information for this task and, as such, plays a major role in the activity of LEACH–SF. The ACO is also used to improve the Sugeno fuzzy rules; this must be done once before implementing the LEACH–SF. This protocol relies on the ABC algorithm for adjusting the fuzzy rules in the LEACH–SF.

**Table 7.** Comparison of studies involving fuzzy hybrid techniques.

| Ref | Algorithm | Optimization and Fuzzy Methods Used | Objective | Optimization Objectives | Fuzzy Input & Output | Defuzzi-fication Method | Fuzzy Rule Evaluations | Matrices | Simula-tion |
|---|---|---|---|---|---|---|---|---|---|
| [89] | GA-ANFIS | Use of ANFIS to group and select CHs in a WSN to ensure low energy usage by the nodes. Harmful nodes in the WSN were discovered by applying a weighted trust evaluation | Increase the lifespan of the WSN | Combines GA and fuzzy logic to detect malicious nodes and extend the WSN's lifetime | Removes the malicious nodes to improve energy usage and lifespan | Center of area | Mamadani | Network lifetime | MATLAB |
| [90] | FAMACROW | Fuzzy logic-based CH selection technique for selecting high-energy nodes and nodes with high-quality communication links and more neighboring nodes as CHs; an ACO-based technique was used for inter-cluster multi-hop routing from CHs to the BS | Remove hotspot problems and reduce energy consumption | Performs unequal clustering to avoid hotspots; performs cluster selection using fuzzy logic, and cluster routing using ACO | Energy levels, number of neighboring nodes, quality of communication link, proficiency of a node to become a CH | Center of area | Mamadani | Energy efficiency, network lifetime | MATLAB |
| [91] | LEACH-SF | The Sugeno fuzzy inference system uses the Fuzzy C-means algorithm to cluster sensor nodes into balanced clusters before selecting the CH. Artificial BCA was used to optimize the fuzzy rules to prolong the service life of the network | Maximize the network lifetime | Applies optimized Sugeno fuzzy system for appropriate CH selection; uses a fuzzy C-means clustering algorithm to form balanced clusters | Energy levels, distance from the BS, distance from the center of gravity, node priority among members of the cluster | Center of area | Sugeno | Data packets received, number of dead nodes, network lifetime | N/A |

## 7. Conclusions

Optimization of the clustering method is typically considered an effective way of achieving optimal energy efficiency in a WSN. A comprehensive review of the recent hierarchical optimization techniques used in cluster head selection, cluster formatting, aggregation, and communication was conducted. The clustering approaches in the available literature were reviewed and classified based on their optimization algorithms. Based on the algorithm type and functioning technique, the protocols were classified into meta-heuristic-based, fuzzy logic-based, and hybrid technique-based. Comparison of the different categories of protocols was based on their performance, clustering and optimization

parameters, and the features that determined their effectiveness. The clustering protocols were compared based on their key features, objectives, and advantages; the protocols were also simulated to compare their ideas and features. The methodology-based comparison of the protocols considered several parameters such as CH parameters, CH rotation, data transmission, method of CH selection, mobility, topology, and deployment policy parameters. This comparison aimed to evaluate the performances of existing clustering methods based on their techniques. The performance-based comparison of the protocols considered protocol type, energy consumption, throughput, stability period, and network lifetime to determine their performances. This review of clustering methods is intended to provide a clear path for future studies in clustered networks.

**Author Contributions:** Supervision: R.H.; validation: A.H.M.A. and H.S.; visualization and writing—original draft: A.M.J.; review and editing: Z.G.A.-M.; B.A.M. and M.S.A. All authors have read and agreed to the published version of the manuscript.

**Funding:** This paper is supported by the Ministry of Higher Education, Malaysia under the Fundamental Research Grant Scheme FRGS/1/2018/TK04/UKM/02/17 and by Universiti Kebangsaan Malaysia (UKM) under the grant Dana Impak Perdana DIP-2018-040.

**Institutional Review Board Statement:** Not applicable.

**Informed Consent Statement:** Not applicable.

**Data Availability Statement:** The data that support the findings of this study are available from the corresponding author, R.H., upon reasonable request.

**Acknowledgments:** The authors would like to acknowledge the support provided by the Network and Communication Technology (NCT) Research Groups, FTSM, UKM in providing facilities throughout this paper.

**Conflicts of Interest:** The authors declare no conflict of interest.

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
