# Peer review of "Optimization of Clustering in Wireless Sensor Networks: Techniques and Protocols"

_applsci, doi:10.3390/app112311448_

Round 1

Reviewer 1 Report

The paper is well written and is acceptable subject to minor language corrections.

This article aims to provide the researchers with an overview of the clustering domain by classifying the approaches into different categories based on optimization algorithms on the basis of their methodologies and functionalities along with the analysis of their limitations and benefits.

As future works, they plan to extend their work into different fields of WSN, for example, Body Area Networks, battery-powered sensor systems, and mobile sink planning. In addition, they also can plan to classify the optimized clustering in terms of WSN types.

Author Response

The manuscript was sent for English proofreading.

Reviewer 2 Report

Dear Authors,

Well done. This is good work to read. However, as this is a review paper, there is an entire section missing in this paper which should be at the first start of the paper to show the following:

  • What are the time limits for this review?  From which year to which year you have searched the literature? For example, you can say that your search is from 2008-2021.
  • What are the keywords for your search?
  • What are the databases that you searched?
  • How do you filter out the papers (references)?

so please refer to the guidelines for producing and writing review papers and try to add a section to explain the above points. 

Away from that, your work is interesting.

Regards 

Author Response

A new section has been added to describe how the review was conducted (page 3, line 107).

Reviewer 3 Report

The presented paper is some kind of survey on techniques and methods for optimization of clustering in wireless sensor networks. The authors prepared an exhaustive and very deep analysis of existing methods, showing the relations and describing the features. 

In my opinion, this paper is very important and should be accepted, despite having no practical parts, no experiments as well as no new solutions. 

Author Response

We highly appreciate your comments which we feel have brought light to our way.